# Determining Priorities in Infrastructure Management Using Multicriteria Decision Analysis

**Ana Bošnjak** [1],*  and **Nikša Jajac** [2]

1  Faculty of Civil Engineering, Architecture and Geodesy, University of Mostar,
   88000 Mostar, Bosnia and Herzegovina
2  Faculty of Civil Engineering, Architecture and Geodesy, University of Split, 21000 Split, Croatia;
   njajac@gradst.hr
*  Correspondence: ana.bosnjak@fgag.sum.ba

**Abstract:** This paper aims to form a concept of infrastructure management based on a multicriteria approach to determining management priorities. As the complexity of infrastructure construction and maintenance management requires looking at this problem from different aspects, the proposed multicriteria approach in this paper is based on the application of a two-phase analytical hierarchy process (AHP) method and technique for order of preference by similarity to ideal solution (TOPSIS) method. Using the two-phase AHP method, the process of determining the relative weights of the criteria is improved with the aim of providing better management of stakeholders as one of the essential preconditions for the success of the entire management process. In this way, it is desired to simulate the decision-making process as realistically as possible, in which the opinions and interests of all stakeholders are respected, but the key decision-maker is responsible for the final decision. Furthermore, with the help of the TOPSIS method, a ranking list of maintenance management priorities is formed, based on which it is possible to distribute limited financial resources intended for annual maintenance more rationally. The stability of the TOPSIS results was confirmed by a sensitivity analysis when changing the relative weights of the criteria. The proposed allocation of financial resources represents the basis for a better design of the maintenance management plan of the analyzed infrastructure elements, thus completing the observed gap in the existing literature. The aim of the above is to improve the planning function and at the same time to improve the implementation, monitoring, and control management functions, which creates a more efficient management system that can preserve the value of the analyzed infrastructure elements and extend their lifetime.

**Keywords:** multicriteria analysis; multicriteria decision-making methods; stakeholders; infrastructure; AHP; TOPSIS

## 1. Introduction

The complexity of infrastructure projects results from their large scope and complex structure, a large amount of required material and human resources, and a large number of stakeholders with different interests and goals, which is why the management of infrastructure construction or maintenance projects often represents a great challenge. According to [1], effective implementation of a project is only possible if the project is realized in an organized manner, whereby the individual activities and processes that make up the whole project are reviewed, coordinated, and directed in detail. Therefore, the implementation of infrastructure projects such as the construction, renovation, or maintenance of railways, roads, pipelines, energy facilities, etc., due to its complexity, dynamism, and uncertainty, necessarily requires management to ensure the required technical performance and quality of the project with minimum time and costs of implementation [1]. Most of such complex projects require the involvement of a large number of stakeholders, their identification, and the assessment of their interests, which is a necessary precondition for the project's

success [2]. According to [3], the authors also point out that the importance of stakeholder engagement is recognized as the key success factor of infrastructure projects. In this case, stakeholders are all persons or organizations that have a positive or negative influence on the outcome of the project. Therefore, stakeholders can be divided into different groups of people organized in different ways (e.g., senior management, users, suppliers, partners, etc.) with different attitudes, interests, and influences. According to the International Project Management Association (IPMA)'s project management standard, one of the fundamental elements of competencies from the field of competencies named "Practice" is exactly the stakeholders. As such, this element of competence with the aim of ensuring successful project management includes the recognition and analysis of the views and expectations of all relevant stakeholders, their engagement, and management [4].

According to [5], due to the complexity and uncertain nature of infrastructure projects, effective stakeholder management approaches are needed to contain conflicting stakeholder interests to build their coexistence during construction and ensure the attainment of the overall organizational goal. Furthermore, according to [6], if the views of the project stakeholders are not addressed, and if the stakeholders are not involved in the development of the project, then the project is unlikely to deliver the optimum value for all involved. In this case, project managers must establish the balance between the stakeholders' involvement and the isolation of the project from external influence to achieve cost-effective and timely project delivery and also maximize the benefit for the client and its stakeholders.

According to [7], multicriteria decision-making (MCDM) is a process of selecting the best option among several alternatives based on multiple criteria that enables the combination and satisfaction of numerous conflicting goals. Therefore, by applying the multicriteria analysis (MCA) and multicriteria decision-making (MCDM) methods in infrastructure management, it is possible to achieve conflicting goals and satisfy the interests of various stakeholders, thus creating a compromise solution when determining management priorities.

As this paper focuses on the maintenance management of selected infrastructure elements according to [8], it is important to recognize the differences in the opinions of involved stakeholders about maintenance priorities so that these differences can be reconciled and thus create a system that meets the needs of all stakeholders. The opinions of all stakeholders are the key information required when determining the relative weights of the criteria and finally determining the priorities of maintenance management using selected MCDM methods. In this way, it is possible to raise the level of transparency and objectivity when choosing a compromise solution. At the same time, it is possible to help top management in making better decisions, for example about investing funds intended for the annual maintenance of a selected infrastructure system based on actual condition. Such decisions are often based solely on the subjective opinion of the individual at the top of the hierarchical structure. However, by applying the MCA, it is possible to obtain a more objective view of the analyzed elements of the infrastructure system. Based on this, it is possible to plan, implement, monitor, and control the maintenance of these elements more effectively during the entire lifetime of the analyzed infrastructure system.

When it comes to the decision-making process with respect to the acceptability of infrastructural solutions according to [9], previously, only economic criteria expressed in monetary measures were taken into account. Such a traditional approach to the valorization of infrastructure solutions was mainly based on the cost analysis of construction, use, and maintenance of the infrastructure and the corresponding benefits. Today, when this type of projects and the conditions of their implementation are more complex, it is necessary to take into account numerous other criteria that can contribute to the sustainable development of the areas to which these projects belong. Different economic, ecological, technical, social, and other criteria that must be taken into account depending on the type of management problem are assessed by different qualitative and quantitative measures. Such criteria can be reduced to a common denominator precisely by applying the multicriteria analysis

(MCA). From the above, the limitation of the mentioned cost–benefit analysis is reducing all criteria exclusively to the monetary value according to certain procedures.

The previously mentioned facts from which the complexity of the management problem arises are common to all types of infrastructure systems. However, when establishing a specific decision support system (DSS), it is necessary to consider the uniqueness of each of them. For example, when it comes to a transport infrastructure system, according to [10], every year traffic accidents lead to the death of almost 1.35 million people worldwide, of which 20% are in developing countries. Furthermore, rapid deterioration of road infrastructure in these countries causes considerable economic losses and puts road safety at risk. The development of road infrastructure, connectivity, accessibility, and mobility are prerequisites for economic growth in developing countries. The usually constrained financial resources intended for road infrastructure maintenance represent a limitation factor and require effective cost management. All the mentioned facts point to the need for an effective maintenance management system for road infrastructure that will enable more rational allocation of funds, extend the life of roads, and prevent premature pavement deterioration.

Therefore, the main goal of this paper is to design a maintenance management concept for the elements of a road infrastructure system, which is based on a multicriteria approach to determining management priorities. The proposed concept can serve as a guideline when creating a priority maintenance plan for a selected road network, while considering the annual maintenance budget. To achieve the main goal, the following research questions were defined:

- Is it possible to establish the concept of maintenance management for an infrastructure system using multicriteria decision-making (MCDM) methods?
- Is it possible to include all stakeholders in the decision-making process concerning the maintenance management of a road infrastructure system by simulating a realistic decision-making process?
- Can the concept of maintenance management based on the application of the MCDM methods be the basis for more rational allocation of limited financial resources intended for the maintenance of a road infrastructure system?

For a better understanding of the described research problem in the continuation of this paper, a review of the relevant literature in the field of application of the DSS based on different MCDM approaches in the management of different types of infrastructure systems and their elements. The Literature Review section is followed by the Description of Methods, Case Study, Discussion, and Conclusions sections.

## 2. Literature Review

To investigate the applicability of the multicriteria decision-making (MCDM) approach on the examples of different infrastructure systems and their elements, a detailed review of the relevant literature is presented in the continuation of this paper.

Badi et al. [11] proposed a hybrid approach to the MCDM integrating grey system theory and the technique for order preference by similarity to the ideal solution (TOPSIS) when choosing the optimal location for the construction of a solar farm. A total of 12 criteria were defined for the evaluation and comparison of the proposed six locations for the construction of solar farms. The grey system theory was used to determine the weights of the defined criteria, while the TOPSIS method was used for the selection of the most suitable location for construction based on the distance from the ideal solution.

Jagtap and Karande [12] selected a nontraditional machining process using an m-polar fuzzy set elimination and choice translating reality-I (ELECTRE-I) approach. The criteria weights for the m-polar fuzzy ELECTRE-I method were evaluated using an analytical hierarchy process (AHP) approach and revised Simos' method. The results revealed that the updated Simos' method, which takes into account user preferences, performs better for the m-polar fuzzy ELECTRE-I algorithm than the AHP weight calculation method.

To improve the quality of building design, Eryuruk et al. [13] proposed an MCDM approach based on the AHP method. The main criteria and subcriteria were determined hierarchically for the determination of mass housing production quality. These criteria were prioritized based on the satisfaction of the stakeholders in the production process [13].

In order to increase the transparency and quality of decision-making, Santa-Cruz et al. [14] presented an approach based on the combination of building information modelling (BIM), information and communication technologies (ICTs), and multicriteria decision-making (MCDM) methods in determining priorities among seismic building renovation projects. The reason for including MCDM is reflected in the possibility of taking into account different criteria such as economic, social, environmental, and other criteria.

Since the maintenance process is a key factor in extending the effective power plan life cycle as well as improving sustainable energy production, Ozcan et al. [15] applied a combination of AHP and TOPSIS methods when choosing a hydropower plant maintenance strategy. As hydroelectric power plants comprise a few thousand pieces of equipment with different characteristics, the most important ones for the maintenance of the analyzed power plants were determined using the TOPSIS method. In this case, the AHP method was used for determining the weights of nine criteria for a big-scale hydroelectric power plant in Turkey.

Munoz-Medina et al. [16] applied an approach also based on a combination of the AHP and TOPSIS methods in the selection of retaining walls. The obtained compromise solution was analyzed by comparing the results with the approach in which the TOPSIS method was replaced by the VIKOR (Višekriterijumska Optimizacija I Kompromisno Rješenje) method.

The implementation of infrastructure projects is associated with a significant degree of uncertainty and insecurity, which can greatly affect the end of the project in terms of contracted deadlines, costs, and quality. Since such projects last a long time and involve a large number of stakeholders through different phases of the project's life cycle, it is difficult to eliminate this uncertainty, but with successful management, it is possible to reduce it. For this purpose, the fuzzy sets theory devised by Zadeh in 1965 is very often applied in the relevant literature. According to that theory, the belonging of an element to a fuzzy set is expressed exclusively by values between 0 and 1.

For example, Sarvari et al. [17] applied a fuzzy analytical hierarchy process (FAHP) method to prioritize contracting methods to determine the most suitable contract option for water and wastewater projects. The results of this study can help the top level of management to choose the best method of contracting different types of projects.

Alfaggi et al. [18] used the FAHP technique to estimate the best roof structure based on the cost ranking among six alternatives of the slab. In this case, the costs of materials, labor, machinery, transportation, and trash on site were all considered. Their research findings show that the models can assist decision-makers in determining the cost rank of the roof selection.

Tamošaitiene et al. [19] used the Delphi technique and the FAHP method to identify and prioritize the criteria for selecting the most appropriate method of repair and maintenance of commercial buildings.

Numerous authors in the relevant literature apply the FAHP method in quantitative research of risk assessment in different areas. For example, in the renovation of buildings, there is often a risk caused by some degree of uncertainty that must be managed. In this case, recognizing a possible danger at the right time is a key factor that can reduce negative impacts on people and the environment. Thus, Hoang [20] applied the FAHP approach to help the stakeholders in the hazard assessment and prediction of possible risks in renovation projects.

Furthermore, Wang et al. [21] presented an MCDM approach based on the TOPSIS method under a picture-fuzzy environment in assessing the risks of a construction project.

In recent research, an increasingly common way of modeling uncertainty is based on intuitionistic fuzzy sets (IFS), which is, according to [22], a very useful approach to reducing the level of uncertainty in the decision-making process. Intuitionistic fuzzy sets represent

an extension of fuzzy sets for the reason that in the fuzzy sets theory the membership of an element to a fuzzy set is represented as a value between 1 and 0 which in reality is often not so simple due to the existence of a certain degree of hesitation in the decision-making process. Because of that, Atanassov, according to [23], proposed the application of intuitionistic fuzzy sets (IFS) that include a degree of hesitation known as the hesitation limit and as such today are useful in numerous areas of application.

For example, Atanassov et al. [24] proposed an intuitionistic fuzzy approach to multicriteria decision-making based on intercriteria analysis. Their research was based on a generalized network (GN) model of the multiexpert multicriteria decision-making process, which was extended by the multicriteria analysis to create a modified set of criteria.

Salimian et al. [25] presented an integration of approaches for obtaining criteria weights and ranking alternatives. Criteria weights were calculated using a combination of the CRITIC method and the ideal point method while alternative ranks are calculated by integrating the evaluation based on distance from average solution (EDAS) and additive ratio assessment (ARAS) approaches under uncertainty. Such an integrated approach was tested on the example of selecting an appropriate sustainable energy project.

When it comes to MCDM approaches related to the planning, construction, and maintenance of road infrastructure, by reviewing the relevant literature, it is possible to see how different MCDM methods have become more applicable and useful in this area in recent times.

For example, Jiang et al. [26] proposed an approach to sustainable urban road planning, taking into account the construction of new roads and the expansion of existing ones to reduce traffic congestion. This functional, economical, and environmentally acceptable approach is based on a digital twin (DT), multicriteria decision-making methods (MCDM) and geographic information system (GIS) called the DT-MCDM-GIS framework.

Kresnanto et al. [27] determined the district road maintenance priorities using the AHP method. Using a pairwise comparison matrix scale, respondents compared five defined criteria with each other, thus determining their relative weights.

Hasnain et al. [28] applied a DSS based on an analytic network process (ANP) when selecting contractors in road construction projects, i.e., in the planning function of such projects.

Furthermore, Siswanto et al. [29] established a DSS based on the AHP method to determine the priorities of road maintenance in Indonesia as a challenge in an attempt to optimize the use of existing resources.

Kilić Pamuković et al. [30] presented a new DSS applicable to the maintenance of damaged pavements. Their system was based on the AHP and PROMETHEE method and as such was tested on the example of the city of Split in Croatia.

Sayadinia and Beheshtinia [31] presented a hybrid MCDM approach for prioritizing road maintenance in Tehran. Their approach was based on a combination of the AHP method for determining the weights of the criteria and the ELECTRE method and Copeland's technique for the final priority ranking.

Hendra et al. [32] proposed an MCDM approach based on a weighted aggregates sum product assessment (WASPAS) method to prioritize road repairs in Parigi Moutong province under a limited rad maintenance budget.

Fawzy et al. [33] applied the AHP method in determining the weights of criteria for prioritizing road maintenance in Egypt.

Based on the presented literature review, in the continuation of this paper, a description of the appropriately selected MCDM methods is provided. After that, a case study on which the proposed MCDM approach was tested is presented as well as a discussion and the main conclusions of this research together with the guidelines for future research.

## 3. Description of Methods

As described earlier through the main goal of this research, this paper proposes the concept of infrastructure maintenance management for road infrastructure systems

whereby the maintenance management is viewed as a phase of a project's life cycle, which consists of the management functions of planning, implementation, monitoring, and control of maintenance. In this paper, the emphasis is placed on improving the quality of decisions related exclusively to the management function of planning.

The application of the MCA aims to define different criteria specifically identified and adapted for the management of selected infrastructure systems. These criteria are expressed in different measurement units and thus affect the determination of the maintenance management priorities. Defining and assessing the importance of the selected criteria requires the inclusion of all stakeholders in the decision-making process. By combining the estimated relative criteria weights and the actual measured values of the criteria for the analyzed roads in this case, it is possible to obtain a maintenance management plan according to the priorities, considering the actual condition in which these roads are found as the elements of the analyzed road infrastructure system. In this way, planning as a basic management function is being improved by creating a quality plan, which makes it possible to reduce deviations from the plan during the implementation of the maintenance activities.

### 3.1. Determining the Relative Weights of Criteria Using the Two-Phase AHP Method

According to [34], the AHP method is a structured technique for analyzing MCDM problems according to a pairwise comparison scale. Using the AHP method, it is possible to determine relative weights of several quantitative or qualitative criteria, taking into account the differences in the experts' opinions and possible conflicts that often occur in reality.

According to the AHP method, unstructured and poorly structured problems are broken down into components such as the main goal, goals, and criteria. After that, these parts are arranged in a hierarchical structure of goals (HSG) for a better understanding of the problem (Figure 1). The number of goals on the second hierarchical level and criteria on the third hierarchical level depends on the type of problem that needs to be solved.

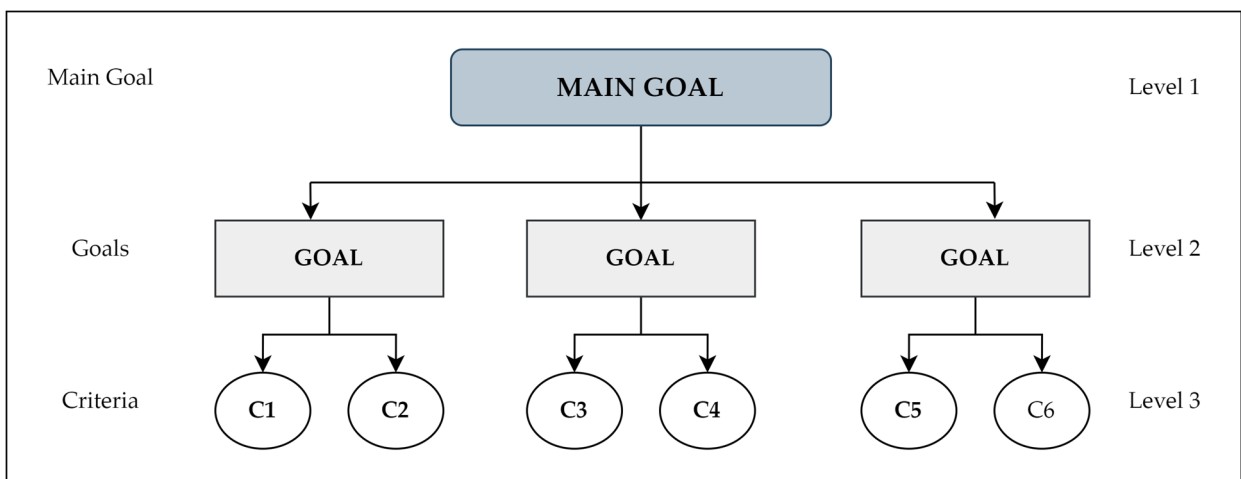

**Figure 1.** Hierarchical structure of goals (HSG).

After the HSG for a given problem has been defined, it is necessary to determine the relative weights of the criteria at the lowest hierarchical level by assigning each criterion a numerical value according to the Saaty's scale of relative importance (Table 1). According to [35], paired comparison judgments in the AHP are applied to pairs of homogeneous elements, and the Saaty's scale has been validated for effectiveness in many applications by several people and through the theoretical justification of what scale one must use in the comparison of homogeneous elements.

**Table 1.** Saaty's fundamental scale [35,36].

| Intensity of Importance | Definition | Explanation |
|:---:|:---:|:---:|
| 1 | Equal importance | Two activities contribute equally to the objective |
| 2 | Weak or slight | |
| 3 | Moderate importance | Experience and judgment slightly favor one activity over another |
| 4 | Moderate plus | |
| 5 | Strong importance | Experience and judgment strongly favor one activity over another |
| 6 | Strong plus | |
| 7 | Very strong or demonstrated importance | An activity is favored very strongly over another with its dominance demonstrated in practice |
| 8 | Very, very strong | |
| 9 | Extreme importance | The evidence favoring one activity over another is of the highest possible order of affirmation |
| Reciprocals of above | If activity i has one of the above nonzero numbers assigned to it when compared with activity j, then j has the reciprocal value when compared with i | A responsible assumption |

According to [35], when comparing two criteria, a score of 1 is assigned, which means that both criteria are equally valuable. As such, these criteria equally contribute to the achievement of the goal at a higher hierarchical level. If one criterion is more important than the other, it is assigned a value of 9. Going from value 1 to value 9, the importance of the first criterion gradually increases to the second criterion, and vice versa.

Based on the obtained estimates, it is necessary to design a decision-making matrix B in which a single element of the matrix $b_{jk}$ represents the importance of the j-th criterion to the k-th criterion. If each element of the decision-making matrix $b_{jk}$ is divided by the sum of all elements of the matrix in the column to which the element belongs, the elements of the normalized decision-making matrix $B_{norm}$ are obtained according to expression (1).

$$\overline{b_{jk}} = \frac{b_{jk}}{\sum\limits_{i=1}^{n} b_{ik}} \tag{1}$$

The relative weight of an individual criterion is obtained as the arithmetic mean of an individual row in the normalized matrix if the elements of the normalized matrix in each row are added and divided by the total number of criteria according to expression (2).

$$w_j = \frac{\sum\limits_{i=1}^{n} \overline{b_{jk}}}{n} \tag{2}$$

One of the conditions of the AHP method is decision consistency, which is expressed by the consistency ratio (CR). In the case of consistent decision-making, the value of CR must not exceed 0.1 or 10%. If the CR is higher than the limit value, the assessment must be repeated. According to [37], the consistency ratio (CR) is calculated as the ratio of the

consistency index (CI) and the random consistency index (RI). Therefore, it is necessary to first calculate the CI as follows:

$$IC = \frac{(\lambda_{\max} - n)}{(n - 1)} \tag{3}$$

where $\lambda_{\max}$ is the largest value of the comparison matrix, and n is the number of criteria that are compared with each other. According to (3), consistency in decision-making depends on the difference between $\lambda_{\max}$ and n. This difference should be greater because in that case, the consistency is also greater.

Finally, it is possible to calculate the required consistency ratio (CR) as follows:

$$CR = \frac{IC}{RI} \tag{4}$$

If the obtained consistency ratio (CR) values are within the allowed limits, the second phase of the AHP method described below follows.

When determining the relative weights of the criteria using the AHP method, all identified stakeholders who influence the analyzed problem can participate, and their opinions are equally important. This is one of the reasons for choosing this method together with its adaptability to the problem of maintenance management of selected infrastructure elements. However, in reality, very often a problem arises whose solution requires the input from some interested participants who do not make the key decision. This case indicates that the opinions of all interested participants are not always respected equally. For example, in the problem of this research, three different groups of stakeholders were identified, which included the population, road managers, and experts in the field of transport and management. All identified stakeholders were included in the decision-making process to express their opinions, but in reality, the key decision on the maintenance management priorities is most often made by a government representative. To simulate this decision-making process as realistically as possible in this research, it was proposed to extend the AHP method in such a way that after determining the relative weights of the criteria in the first phase, the second phase of the AHP method is performed. After determining the relative weights of the criteria in the second phase of the AHP method, the key stakeholder performed a separate comparison of the objectives at the second hierarchical level to determine the relative weights of the goals. After that, the relative weights of the criteria from the first phase of the AHP method were adjusted to the relative weights of the goals on the second hierarchical level according to the following expression:

$$w_{c_{ij}} = w_{g_i} * w_{c_i, i=1,...,n; j=1,...,m} \tag{5}$$

where $w_{cij}$ is the final relative weight of the i-th criterion belonging to the j-th goal. For that matter, $w_{cij}$ is calculated as the product of the weight of the i-th goal determined by the representative of the government and the weight of the i-th criterion obtained according to the assessment of all involved stakeholders. In this way, it is possible to include the opinions of all stakeholders in the decision-making process, which is a prerequisite for successful planning, but the final decisions at the strategic level are made in this case by government representatives, thus deciding on the future development of the analyzed infrastructural elements. According to that, the decision-making process is simulated from the operational and tactical levels to the strategic decision-making or management level. At the same time, lower hierarchical levels provide information about the opinions of the involved stakeholders through information flows to higher levels of decision-making and management [38].

After the adjustment, the final relative weights of the criteria from the second phase of the AHP method represents the input data for determining the maintenance management priorities, using the TOPSIS method, which will be described below.

*3.2. Final Ranking of Alternative Solutions Using the TOPSIS Method*

According to [39], the TOPSIS method is one of the useful MCDM techniques to manage real-world problems. This method was proposed by Hwang and Joon, and it is based on the idea that the alternative solution that has the smallest distance from the positive ideal solution, and the largest distance from the negative ideal solution within the Euclidean space is considered the best [40,41]. In this paper, the TOPSIS method was used for the final ranking of alternatives in the road infrastructure maintenance management, which was carried out through several steps shown below according to [42–44].

In the first step, it is necessary to form a normalized decision matrix ($r_{jk}$) by dividing each element of the initial matrix, i.e., the values of the defined criteria for individual alternative solutions, by the total sum of the elements in the corresponding column as follows:

$$r_{jk} = \frac{w_{jk}}{\sqrt{\sum\limits_{k=1}^{n} x_{jk^2}}}, k = 1, \dots n \tag{6}$$

In the second step, it is necessary to multiply the elements of the normalized matrix by the previously obtained relative weights of the criteria as follows:

$$v_{jk} = w_j * r_{jk} \tag{7}$$

In the third step, it is necessary to determine the positive ideal solution (PIS) and the negative ideal solution (NIS) as follows:

$$A^* = \left\{v_1^*, v_2^*, \dots, v_n^*\right\}$$
$$A^- = \left\{v_1^-, v_2^-, \dots, v_n^-\right\} \tag{8}$$

In the fourth step, it is necessary to calculate the Euclidean distances from the positive ideal solution (PIS) and the negative ideal solution (NIS) according to expression (6) as follows:

$$d_i^* = \sqrt{\sum\limits_{j=1}^{n} \left(v_{ij} - v_j^*\right)^2}$$
$$d_i^- = \sqrt{\sum\limits_{j=1}^{n} \left(v_{ij} - v_j^-\right)^2} \tag{9}$$

In the last, sixth, step, it is necessary to calculate the relative distance from the ideal solution ($C_i$) as follows:

$$C_i = \frac{d_i^-}{d_i^- + d_i^*}, i = 1, 2, \dots, n \tag{10}$$

According to expression (10), the $C_i$ value is closer to 1, the alternative solution is closer to the ideal solution, while those solutions for which this value is 0 represent negative ideal solutions. Based on the value of the relative distance from the ideal solution ($C_i$), it is possible to form a list of management priorities.

In the continuation of this paper, the described methodology is applied to the case study of determining the maintenance management priorities for the road infrastructure elements, whereby the HSG is first defined by involving all stakeholders. Based on the defined HSG, the relative weights of the criteria are determined by applying the two-phase AHP method. After that, using the TOPSIS method, a maintenance management plan is generated, taking into account the amount of limited financial resources intended for regular maintenance.

## 4. Case Study

According to [9], investment in infrastructure and its development is an integral part of the expansion, reconstruction, and replacement of existing, outdated infrastructure systems and facilities. A construction infrastructure system as a whole consists of transport, water

supply, energy, and telecommunications infrastructure systems, and for the development of an economically significant region, each of them needs to be managed effectively.

In the continuation of the paper, the results of the application of the described management concept with the MCDM methods are presented using the example of the maintenance management of a road infrastructure system. By observing this problem as a poorly structured problem, the best variant solution is sought to the defined criteria and their relative weights to improve the quality of decision-making in the planning of transport infrastructure elements.

The proposed multicriteria approach to the maintenance management based on a combination of the two-phase AHP method and the TOPSIS method in this case study was applied to the real network of regional roads in Bosnia and Herzegovina. Bosnia and Herzegovina like most other developing countries allocates limited financial resources for regular road maintenance, while minor attention is paid to determining management priorities for more efficient allocation of these resources. For this reason, the improvement of cost management, stakeholder management, and the entire maintenance management process of the analyzed roads is one of the goals of this paper. The proposed approach to solving this problem is shown in Figure 2.

The previously presented maintenance management process of the analyzed infrastructure elements begins with defining the subject and scope of this research and the alternatives. The scope of this research includes a total of twelve regional roads of Herzegovina-Neretva County located in the southern part of Bosnia and Herzegovina. According to the number of inhabitants, it is the eighth most closely populated county in Bosnia and Herzegovina, and its strategic position is very advantageous for connection and further development of the country. Table 2 shows specific data on the analyzed road sections such as length, width, year of construction, or year of the last road rehabilitation and the expert assessment of road condition. The expert assessment of the condition of individual regional roads was obtained as the arithmetic mean of the grades from 1 to 10 according to the maintenance manager's opinion. The assessment was made in such a way that the road with a rating of 1 was in the best condition, while the road with a rating of 10 was in the worst condition.

The next step of the proposed multicriteria approach is the identification of the involved stakeholders and the definition of the hierarchical structure of goals (HSG). By analyzing the interests and influence of individual stakeholders, an HSG is defined, and in this case, it consists of three hierarchical levels. At the first hierarchical level, there is a sustainable maintenance management system of infrastructure elements as the main goal (MG) that needs to be achieved. The main goal (MG) is broken down into three goals (G1, G2, G3) at the second hierarchical level, and each of them is further broken down into a total of nine technical, economic, and social criteria at the last hierarchical level (Table 2). Since the problem of the maintenance management of the road infrastructure system elements is a poorly structured problem according to the opinions of the involved stakeholders, it is necessary to analyze it from the technical, economic, and social aspects. For this reason, the identified stakeholders are divided into three expert groups, among which the first expert group (EG1) includes the users of the analyzed roads, i.e., residents who live in the areas through which the analyzed roads pass, the second expert group (EG2) includes selected scientists and experts from the field of road construction and project management, while the third expert group (EG3) consists of road managers, i.e., persons responsible for the maintenance management process. The selected respondents form a heterogeneous group to look at the problem from different viewing angles. In this case, the users look at the problem from the aspect of road functionality and the impact on driving comfort and ease of use. The experts on road construction look at this problem from the aspect of satisfying the technical conditions of the road. The experts on project management look at the broader picture of this problem, while the maintenance managers look at the problem most often from the strategic aspect and the aspect of limited financial resources. Because of that, each of the expert groups define the criteria that, from their perspective on the problem, they

consider to be significant for determining the maintenance management priorities. All the described elements of the hierarchical structure of goals are presented in detail in Table 3.

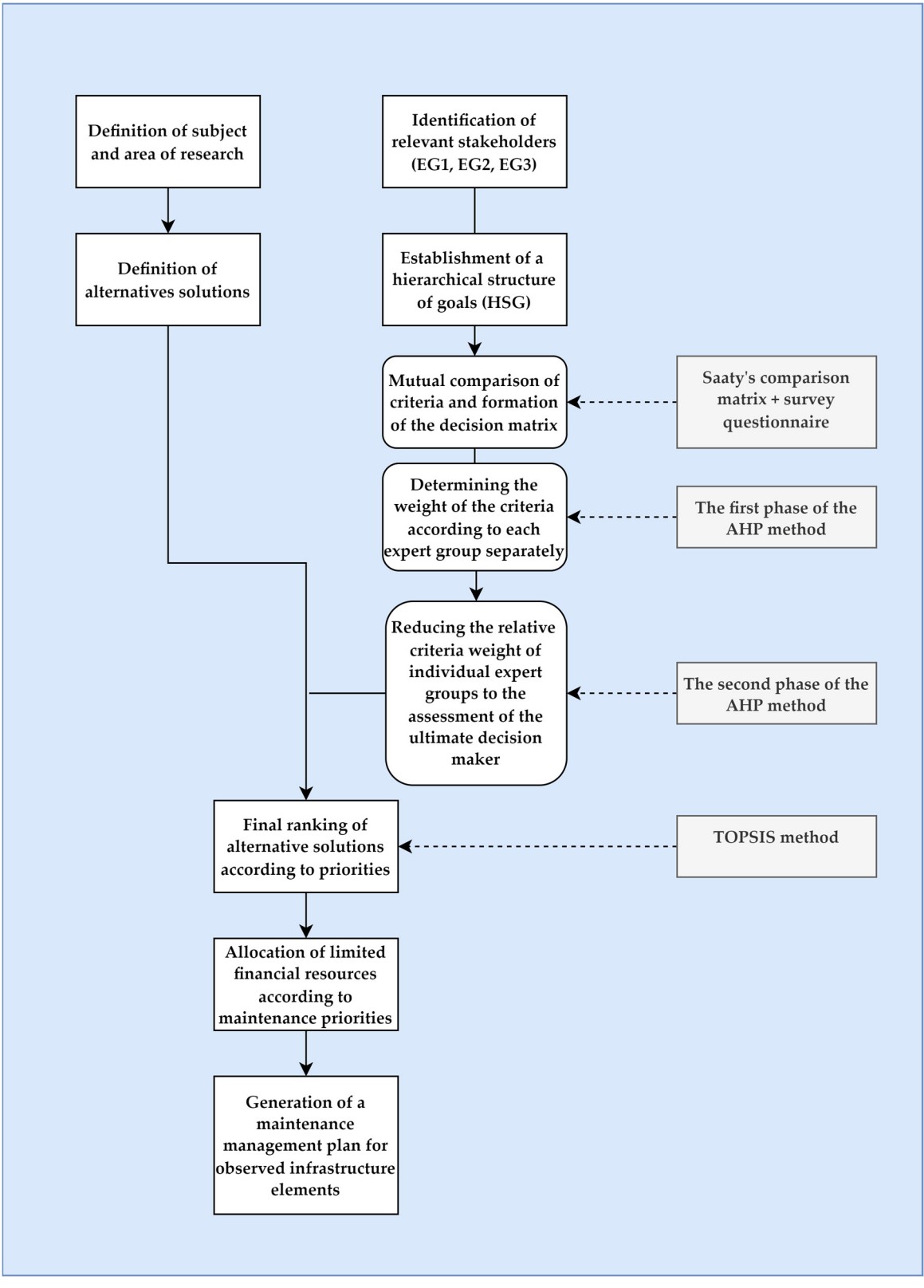

**Figure 2.** The maintenance management concept of the analyzed infrastructure elements.

**Table 2.** General information about the analyzed alternative solutions.

| Alternative | Label | Length | Width | Construction or Rehabilitation Year | Expert Assessment of Road Condition |
|---|---|---|---|---|---|
| section 1 | A1 | 24.67 km | 5.5 m | 2011 | 6 |
| section 2 | A2 | 62.61 km | 4.8 m | 2010 | 3.33 |
| section 3 | A3 | 30.67 km | 5.2 m | 2002 | 3 |
| section 4 | A4 | 23.56 km | 5.0 m | 1983 | 4.67 |
| section 5 | A5 | 18.95 km | 5.0 m | 1985 | 6.33 |
| section 6 | A6 | 43.93 km | 2.7 m | 2000 | 3 |
| section 7 | A7 | 6.52 km | 3.0 m | 1979 | 4 |
| section 8 | A8 | 17.68 km | 4.0 m | 2011 | 4.67 |
| section 9 | A9 | 33.55 km | 3.6 m | 1997 | 3 |
| section 10 | A10 | 24.00 km | 4.6 m | 2011 | 3 |
| section 11 | A11 | 45.66 km | 4.0 m | 1989 | 3 |
| section 12 | A12 | 32.81 km | 3.0 m | 2007 | 2.67 |

**Table 3.** Hierarchical structure of goals (HSG).

| Level of HSG | Element of HSG (MG/G/C) | Name of the HSG Element | Detailed Description of HSG Elements |
|---|---|---|---|
| 1 | Main goal (MG) | A sustainable maintenance management system of infrastructure elements | Establishment of an effective maintenance management system as a basis for planning function |
| 2 | Goal (G1) | Preservation of the technical value of infrastructure elements | Viewing the problem from the technical aspect |
| 2 | Goal (G2) | Preservation of the economic value of infrastructure elements | Viewing the problem from the economic aspect |
| 2 | Goal (G3) | Preservation of the social value of infrastructure elements | Viewing the problem from the social aspect |
| 3 | Technical criterion (TC1) | Traffic intensity | Traffic intensity refers to the concentration of vehicles within the analyzed road at a certain time. It is expressed in the vehicle measurement unit per day. |
| 3 | Technical criterion (TC2) | Road width | Road width is the distance between road boundaries including footpaths and drains expressed in meters. The width of the road is viewed from the aspect of safety and the need for increased maintenance activities. |
| 3 | Technical criterion (TC3) | Type of pavement | The material from which the pavement is made. Most of the analyzed roads have macadam or asphalt pavement. |
| 3 | Technical criterion (TC4) | Share of heavy vehicles | The share of heavy vehicles is expressed as the amount of heavy vehicles in the average annual daily traffic on the analyzed road. |
| 3 | Technical criterion (TC5) | Expert assessment of road conditions | The assessment of the overall condition from 1 to 10 according to the manager who is responsible for the maintenance of the analyzed road. |
| 3 | Economic criterion (EC1) | Maintenance costs | The amount of financial resources that are issued annually for the maintenance of the analyzed road. |

**Table 3.** *Cont.*

| Level of HSG | Element of HSG (MG/G/C) | Name of the HSG Element | Detailed Description of HSG Elements |
|---|---|---|---|
| 3 | Economic criterion (EC2) | Economic significance of the analyzed roads | The impact of the analyzed road on the economic development of the country as a whole. |
| 3 | Social criterion (SC1) | Social significance of analyzed roads | Social significance is expressed through the population and the number of inhabitants in the area through which the analyzed road passes. |
| 3 | Social criterion (SC2) | Time of construction or the last reconstruction of the road | The general condition of the road is significantly affected by the time of construction or the last rehabilitation of the road. |

According to [38], a DSS for infrastructure management consists of three levels of decision-making, namely the operational, tactical, and strategic decision-making levels. The first decision-making level provides support to decision-makers at the operational decision-making level and provides information to higher levels of decision-making through information flows. In this case, the first, operational decision-making level corresponds to the first expert group (EG1). At the next, tactical, level, individual experts and expert teams make decisions, which in this case corresponds to the second expert group (EG2). With the help of the second expert group (EG2)'s opinion and taking into account the opinion of the first expert group (EG1), managers who belong to the third expert group ultimately make decisions on the development of the analyzed infrastructure system at the highest, strategic, level of management. In other words, a lower hierarchical level helps a higher hierarchical level to make the best decision. The connection between the decision-making levels and the associated expert groups of stakeholders in the decision-making process is shown in Figure 3.

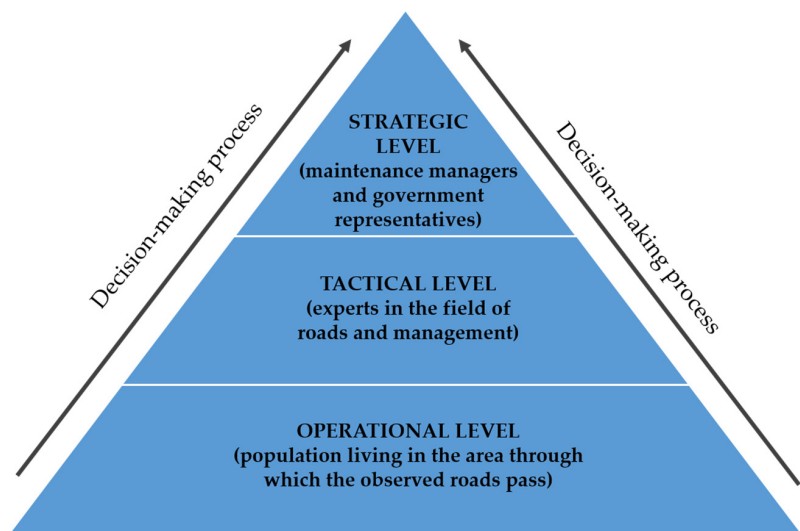

**Figure 3.** Decision-making process according to hierarchical levels.

After defining the HSG, the stakeholders, using the two-phase AHP method, determine the final relative weights of the criteria at the lowest level of the hierarchical structure of goals.

In the first phase of the AHP method, each expert group determined the relative weights of those criteria that were appropriate for the scope of that expert group. Therefore, the users who belonged to EG1 determined only the weights of the social criteria, while the experts on road construction and project management (EG2) and the maintenance managers (EG3) determined the weights of the technical and economic criteria. A total

of 15 respondents made a mutual comparison of the criteria using a prepared survey questionnaire. In the total number of respondents, there were five respondents in each expert group. In the first expert group (EG1), all respondents were the residents of the places through which the analyzed regional roads pass. The members of EG2 were experts from the University of Mostar and the University of Split, while EG3 consisted of four representatives of the Ministry of Transport and Communications of Herzegovina-Neretva County and the Minister of Transport and Communications as the ultimate decision-maker with respect to the maintenance management priorities. The respondents' answers were used as the input data for forming the initial decision-making matrix when determining the relative weights of individual criteria.

The first expert group (EG1) compared the social criteria (SC1, SC2) with each other using the Saaty's scale of relative importance according to the previously described procedure. Based on the obtained data, an initial decision matrix was formed, which was then normalized in such a way that each element of the initial matrix was divided by the sum of all elements of the initial matrix in the corresponding column. Furthermore, the initial relative weight of each criterion was calculated in such a way that the elements of the normalized matrix in the corresponding row were summed and divided by the total number of criteria. After the relative weights of the social criteria were determined, and the consistency ratio was checked in the second phase of the AHP method, it was necessary to multiply the obtained initial relative weights by the relative weight of the goal "preservation of the social value of infrastructure elements" (G3) at the second hierarchical level. In this way, the users' opinion was included in the decision-making by the government representative, which simulated the real picture of the decision-making process where the users can express their views and opinions that influence the decision of the ultimate decision-maker. The described procedure of the two-phase AHP method is shown in detail in Figure 4.

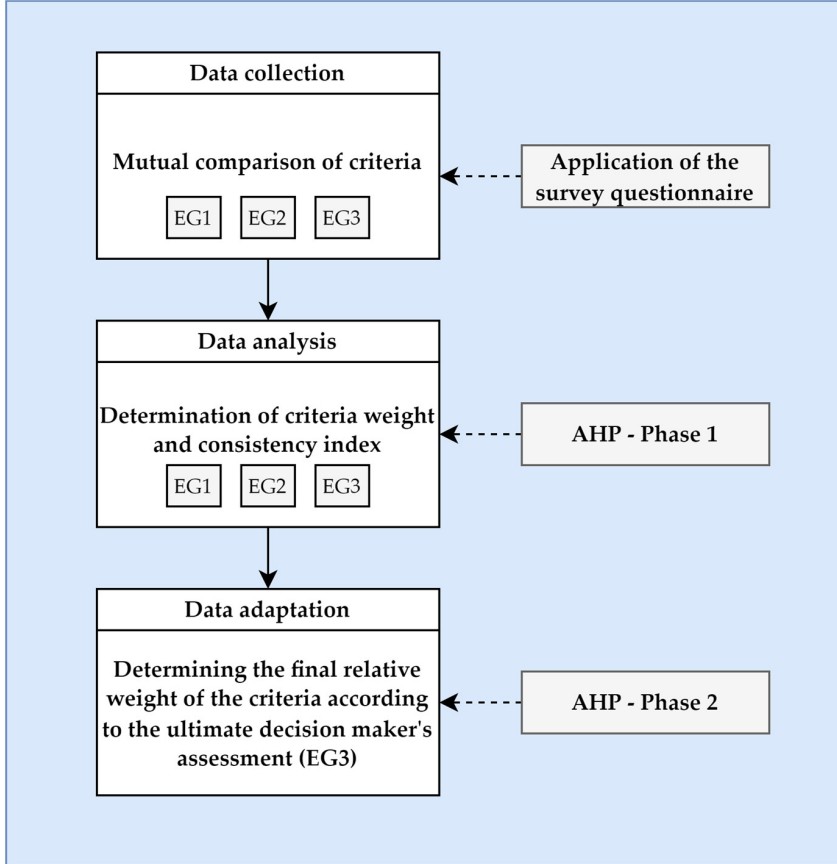

**Figure 4.** Relative weights of the criteria using the two-phase AHP method.

Table 4 shows the obtained relative weights of the social criteria for both phases of the AHP method according to the five stakeholders comprising EG1. According to the results of the second phase of the AHP method, the final relative weight of criterion SC1 is 0.245, and the final relative weight of criterion SC2 is 0.058.

**Table 4.** Relative weights of social criteria—EG1.

| Label of Stakeholder | AHP Phase 1 | | AHP Phase 2 | |
|---|---|---|---|---|
| | SC1 | SC2 | SC1 | SC2 |
| SH1 | 0.833 | 0.167 | 0.252 | 0.051 |
| SH2 | 0.875 | 0.125 | 0.265 | 0.038 |
| SH3 | 0.750 | 0.250 | 0.227 | 0.076 |
| SH4 | 0.833 | 0.167 | 0.252 | 0.051 |
| SH5 | 0.750 | 0.250 | 0.227 | 0.076 |
| SUM | 0.808 | 0.192 | 0.245 | 0.058 |

Similarly, the second expert group (EG2) performed a mutual comparison of the technical (TC1, TC2, TC3, TC4, TC5) and economic criteria (EC1, EC2) defined according to the procedure and rules of the regular maintenance of the analyzed roads. Based on the assessment made using the Saaty's scale of relative importance, the previously described procedure was repeated in such a way as to form an initial matrix and a normalized decision matrix, from which the initial relative weights of the technical and economic criteria were determined in the first phase of the AHP method. According to the last line of Table 5, it can be seen that the EC2 criterion has the highest initial relative weight of 0.642 in the first phase of the AHP method, and the TC2 criterion has the lowest initial relative weight of 0.058.

**Table 5.** Relative weights of technical and economic criteria in the AHP Phase 1—EG2.

| Label of Stakeholder | AHP Phase 1 | | | | | | |
|---|---|---|---|---|---|---|---|
| | TC1 | TC2 | TC3 | TC4 | TC5 | EC1 | EC2 |
| SH6 | 0.210 | 0.098 | 0.088 | 0.105 | 0.499 | 0.250 | 0.750 |
| SH7 | 0.577 | 0.033 | 0.110 | 0.124 | 0.157 | 0.750 | 0.250 |
| SH8 | 0.180 | 0.039 | 0.139 | 0.217 | 0.424 | 0.167 | 0.833 |
| SH9 | 0.154 | 0.042 | 0.041 | 0.199 | 0.564 | 0.500 | 0.500 |
| SH10 | 0.201 | 0.080 | 0.034 | 0.513 | 0.173 | 0.125 | 0.875 |
| SUM | 0.264 | 0.058 | 0.082 | 0.232 | 0.363 | 0.358 | 0.642 |

The initial relative weights of the technical and economic criteria in the second phase of the AHP method were also adjusted to the evaluation of the goals (G1, G2) by the representative of the government (Table 6).

**Table 6.** Relative weights of technical and economic criteria in the AHP Phase 2—EG2.

| Label of Stakeholder | AHP Phase 2 | | | | | | |
|---|---|---|---|---|---|---|---|
| | TC1 | TC2 | TC3 | TC4 | TC5 | E1 | E2 |
| SH6 | 0.127 | 0.059 | 0.053 | 0.064 | 0.303 | 0.023 | 0.068 |
| SH7 | 0.350 | 0.020 | 0.067 | 0.075 | 0.095 | 0.068 | 0.023 |
| SH8 | 0.109 | 0.024 | 0.084 | 0.132 | 0.257 | 0.015 | 0.075 |
| SH9 | 0.093 | 0.025 | 0.025 | 0.121 | 0.342 | 0.045 | 0.045 |
| SH10 | 0.122 | 0.049 | 0.021 | 0.311 | 0.105 | 0.011 | 0.079 |
| SUM | 0.160 | 0.035 | 0.050 | 0.141 | 0.221 | 0.032 | 0.058 |

According to the last line of Table 6, it can be seen that the most important criterion with the greatest final relative weight is criterion TC1, and the least important criterion is

EC1. It is evident from this that the involved stakeholders had the opportunity to express their opinion, but the final decision was made by the representative of the government who also belonged to the third expert group (EG3).

Furthermore, Table 7 shows the initial relative weights of the technical and economic criteria according to EG3 which included five maintenance managers of the analyzed roads for the first phase of the AHP method. The initial relative weights of the criteria were determined using the same procedure of the AHP method as the one used by the remaining expert groups. Of the technical criteria, TC2 had the greatest initial relative weight, while EC1 was the most important of the economic criteria.

**Table 7.** Relative weights of technical and economic criteria in the AHP Phase 1—EG3.

| Label of Stakeholder | AHP Phase 1 | | | | | | |
|---|---|---|---|---|---|---|---|
| | TC1 | TC2 | TC3 | TC4 | TC5 | EC1 | EC2 |
| SH11 | 0.249 | 0.311 | 0.105 | 0.235 | 0.063 | 0.500 | 0.500 |
| SH12 | 0.230 | 0.369 | 0.092 | 0.257 | 0.052 | 0.500 | 0.500 |
| SH13 | 0.382 | 0.247 | 0.049 | 0.219 | 0.103 | 0.500 | 0.500 |
| SH14 | 0.228 | 0.334 | 0.088 | 0.242 | 0.048 | 0.667 | 0.333 |
| SH15 | 0.279 | 0.376 | 0.123 | 0.172 | 0.049 | 0.750 | 0.250 |
| SUM | 0.274 | 0.335 | 0.091 | 0.225 | 0.063 | 0.583 | 0.417 |

The data from Table 7 were adapted to the assessment of the elements at a higher level of the HSG again, and the final relative weights of the criteria according to this group were obtained in the second phase of the AHP method (Table 8). From the last row of Table 6, it is possible to see that among the technical criteria, TC2 still has the greatest relative importance, while EC1 is the most important among the economic criteria.

**Table 8.** Relative weights of technical and economic criteria in the AHP Phase 2—EG3.

| Label of Stakeholder | AHP Phase 2 | | | | | | |
|---|---|---|---|---|---|---|---|
| | TC1 | TC2 | TC3 | TC4 | TC5 | EC1 | EC2 |
| SH11 | 0.151 | 0.211 | 0.064 | 0.143 | 0.038 | 0.045 | 0.045 |
| SH12 | 0.140 | 0.224 | 0.056 | 0.156 | 0.032 | 0.045 | 0.045 |
| SH13 | 0.232 | 0.150 | 0.030 | 0.133 | 0.063 | 0.045 | 0.045 |
| SH14 | 0.138 | 0.203 | 0.053 | 0.147 | 0.029 | 0.060 | 0.030 |
| SH15 | 0.169 | 0.228 | 0.075 | 0.104 | 0.030 | 0.068 | 0.023 |
| SUM | 0.166 | 0.203 | 0.055 | 0.137 | 0.038 | 0.053 | 0.037 |

From all previously presented data according to the second phase of the AHP method, Table 9 gives a summary of the relative weights of the criteria according to all interested participants. The final relative weights of the technical and economic criteria were obtained as the arithmetic mean of the final relative weights of these criteria according to EG2 and EG3, while the final relative weights of the social criteria were obtained according to EG1. Furthermore, the sum of the final relative weights of the technical criteria corresponds to the relative weight of G1 at the higher level of the HSG which is 0.607. Accordingly, the sum of the final relative weights of the economic criteria corresponds to the relative weight of G2 which is 0.090, while the sum of the final relative weights of the social criteria corresponds to the relative weight of G3 which is 0.303 (Table 9).

The final relative weights of the criteria obtained by applying the two-phase AHP method were used as the input data for the TOPSIS method according to which the aim was to rank the alternatives for the HSG. In this case study, they included twelve regional roads in the area of Herzegovina-Neretva County in Bosnia and Herzegovina according to the priorities of their maintenance with respect to quantitative and qualitative criteria expressed in different measurement units.

**Table 9.** Summary of the relative weights of criteria.

| Criteria | Relative Weight—EG1 | Relative Weight—EG2 | Relative Weight—EG3 | Criterion Weight on the 3rd Level of HSG | Goal Weight on the 2nd Level of HSG | |
|---|---|---|---|---|---|---|
| TC1 | | 0.160 | 0.166 | 0.163 | | |
| TC2 | | 0.035 | 0.203 | 0.119 | | |
| TC3 | | 0.050 | 0.055 | 0.053 | G1 | 0.607 |
| TC4 | | 0.141 | 0.137 | 0.139 | | |
| TC5 | | 0.221 | 0.038 | 0.129 | | |
| EC1 | | 0.032 | 0.053 | 0.042 | G2 | 0.090 |
| EC2 | | 0.058 | 0.037 | 0.048 | | |
| SC1 | 0.245 | | | 0.245 | G3 | 0.303 |
| SC2 | 0.058 | | | 0.058 | | |

Table 10 shows the weighted decision matrix for all twelve sections of the analyzed regional road network. The elements of the weighted matrix were obtained by multiplying the values of individual criteria for the analyzed roads by the final relative weights of the criteria from Table 9.

**Table 10.** Weighted decision matrix.

| Criterion/Alternative | Label | TC1 | TC2 | TC3 | TC4 | TC5 | EC1 | EC2 | SC1 | SC2 |
|---|---|---|---|---|---|---|---|---|---|---|
| $w_j$ | - | 0.613 | 0.119 | 0.053 | 0.139 | 0.129 | 0.042 | 0.048 | 0.245 | 0.058 |
| section 1 | A1 | 0.316 | 0.119 | 0.053 | 0.139 | 0.122 | 0.010 | 0.028 | 0.243 | 0.058 |
| section 2 | A2 | 0.122 | 0.105 | 0.026 | 0.059 | 0.068 | 0.006 | 0.032 | 0.245 | 0.058 |
| section 3 | A3 | 0.074 | 0.112 | 0.051 | 0.013 | 0.061 | 0.011 | 0.024 | 0.135 | 0.058 |
| section 4 | A4 | 0.385 | 0.117 | 0.053 | 0.023 | 0.095 | 0.015 | 0.048 | 0.116 | 0.057 |
| section 5 | A5 | 0.613 | 0.107 | 0.053 | 0.033 | 0.129 | 0.018 | 0.044 | 0.093 | 0.057 |
| section 6 | A6 | 0.081 | 0.060 | 0.053 | 0.084 | 0.061 | 0.008 | 0.040 | 0.003 | 0.058 |
| section 7 | A7 | 0.166 | 0.064 | 0.053 | 0.011 | 0.081 | 0.042 | 0.016 | 0.240 | 0.058 |
| section 8 | A8 | 0.071 | 0.092 | 0.053 | 0.035 | 0.095 | 0.019 | 0.036 | 0.051 | 0.058 |
| section 9 | A9 | 0.281 | 0.068 | 0.028 | 0.077 | 0.061 | 0.010 | 0.020 | 0.131 | 0.058 |
| section 10 | A10 | 0.112 | 0.109 | 0.030 | 0.009 | 0.061 | 0.013 | 0.012 | 0.028 | 0.058 |
| section 11 | A11 | 0.067 | 0.091 | 0.041 | 0.018 | 0.061 | 0.007 | 0.004 | 0.179 | 0.057 |
| section 12 | A12 | 0.054 | 0.075 | 0.026 | 0.022 | 0.054 | 0.007 | 0.008 | 0.166 | 0.058 |
| MIN | - | 0.054 | 0.060 | 0.026 | 0.009 | 0.054 | 0.006 | 0.004 | 0.003 | 0.057 |
| MAX | - | 0.613 | 0.119 | 0.053 | 0.139 | 0.129 | 0.042 | 0.048 | 0.245 | 0.058 |

According to the previously described procedure of the TOPSIS method, Table 11 shows the final ranking of the analyzed alternatives according to the maintenance management priorities. The ranking of the priorities was obtained by calculating the Euclidean distances from the positive ideal solution ($d_{i*}$) and the negative ideal solution ($d_{i-}$) according to expression (6). After that, expression (7) was used to obtain the relative distance from the ideal solution ($C_i$). According to Table 11, section 5 with the $C_i$ value of 0.755 is the closest to value 1 or to the ideal solution, which is why this alternative is ranked as best. Going from section 5 to section 10, the relative distance from the ideal solution ($C_i$) gradually decreases, thus determining the priorities of the maintenance management for the analyzed roads as the elements of the analyzed infrastructure system. In this case, section 5 requires the lowest level of regular maintenance activities, while section 10 requires the highest level of regular maintenance activities for the analyzed roads to be in a satisfactory condition and provide necessary services for their users.

After the ranking of priorities is determined by applying the TOPSIS method, it is necessary to perform a sensitivity analysis to determine the impact of the change in the input data on the obtained ranking list of the maintenance management priorities. The sensitivity analysis indicates the reliability of the selected ranking method and is performed by changing the final relative weights of the criteria. Changing the weights of the criteria is performed in such a way that the relative weight of one criterion increases by a value from

0.1 to 0.9 in each iteration, while the values of the other criteria are fixed and unchanging. The results for all alternatives and all scenarios obtained by gradually increasing the final relative weight of one criterion to other criteria are shown in Table 12.

**Table 11.** Complete ranking of alternatives according to the TOPSIS method.

| Alternative | Label | $d_{i*}$ | $d_{i-}$ | $C_i$ | Rank of Alternative |
|---|---|---|---|---|---|
| section 1 | A1 | 0.300 | 0.390 | 0.565 | 2 |
| section 2 | A2 | 0.503 | 0.262 | 0.342 | 6 |
| section 3 | A3 | 0.570 | 0.147 | 0.206 | 9 |
| section 4 | A4 | 0.290 | 0.360 | 0.554 | 3 |
| section 5 | A5 | 0.187 | 0.575 | 0.755 | 1 |
| section 6 | A6 | 0.595 | 0.091 | 0.133 | 10 |
| section 7 | A7 | 0.472 | 0.268 | 0.362 | 5 |
| section 8 | A8 | 0.587 | 0.089 | 0.132 | 11 |
| section 9 | A9 | 0.370 | 0.270 | 0.422 | 4 |
| section 10 | A10 | 0.567 | 0.081 | 0.125 | 12 |
| section 11 | A11 | 0.571 | 0.180 | 0.239 | 7 |
| section 12 | A12 | 0.586 | 0.164 | 0.219 | 8 |

**Table 12.** Results of sensitivity analysis.

| Ø | A1 | A2 | A3 | A4 | A5 | A6 | A7 | A8 | A9 | A10 | A11 | A12 |
|---|---|---|---|---|---|---|---|---|---|---|---|---|
| 0.1 | 0.565 | 0.342 | 0.206 | 0.554 | 0.75438 | 0.133 | 0.362 | 0.132 | 0.422 | 0.125 | 0.239 | 0.219 |
| 0.2 | 0.566 | 0.343 | 0.206 | 0.554 | 0.75402 | 0.133 | 0.363 | 0.132 | 0.422 | 0.125 | 0.240 | 0.219 |
| 0.3 | 0.567 | 0.343 | 0.206 | 0.554 | 0.75347 | 0.134 | 0.363 | 0.133 | 0.422 | 0.125 | 0.240 | 0.219 |
| 0.4 | 0.567 | 0.344 | 0.207 | 0.553 | 0.75278 | 0.134 | 0.363 | 0.133 | 0.422 | 0.125 | 0.240 | 0.220 |
| 0.5 | 0.568 | 0.344 | 0.207 | 0.553 | 0.75197 | 0.135 | 0.364 | 0.134 | 0.422 | 0.125 | 0.241 | 0.220 |
| 0.6 | 0.569 | 0.345 | 0.208 | 0.552 | 0.75109 | 0.135 | 0.364 | 0.135 | 0.422 | 0.125 | 0.241 | 0.220 |
| 0.7 | 0.570 | 0.345 | 0.209 | 0.552 | 0.75013 | 0.136 | 0.365 | 0.136 | 0.423 | 0.125 | 0.242 | 0.221 |
| 0.8 | 0.571 | 0.346 | 0.209 | 0.551 | 0.74913 | 0.136 | 0.365 | 0.137 | 0.423 | 0.126 | 0.242 | 0.221 |
| 0.9 | 0.573 | 0.347 | 0.210 | 0.551 | 0.74809 | 0.137 | 0.366 | 0.138 | 0.423 | 0.126 | 0.242 | 0.221 |

Based on the previously presented data, it can be seen that the ranking of the alternative solutions does not change by changing the final relative criterion weights, which indicates the stability of the obtained results (Figure 5).

Material resources intended for the maintenance management of the analyzed infrastructure elements are regularly limited. Therefore, the previously obtained priority ranking based on the actual state of the analyzed roads and expressed through the measured values of the defined criteria helps to properly distribute financial resources for the real needs. In this way, each of the analyzed roads with a sufficient amount of allocated financial resources could retain its technical, economic, and social value by serving its purpose.

Table 13 shows the proposed allocation of the annual budget intended for the maintenance management of the analyzed roads based on the priority ranking obtained by applying the two-phase AHP and TOPSIS methods. These data represent the basis for creating an annual maintenance management plan for the analyzed roads, which can be used to improve the other management functions of implementation, monitoring, and control in the overall management process. In Table 11, section 5 as the first-rank alternative is in the best condition and accordingly requires the least amount of financial resources to maintain this satisfactory condition. Therefore, Table 13 shows that only 2.67% of the total annual budget for these roads, which amounts to EUR 765,000.00, needs to be set aside for the regular maintenance of this road. Furthermore, according to the results of the TOPIS method, section 10 is in the worst condition. For this regional road, it is necessary to allocate 16.15% of the total annual budget, which amounts to EUR 123,572.58. The remaining sections are between these two extremes, and the sum of all percentages is 100% which corresponds to the total annual budget.

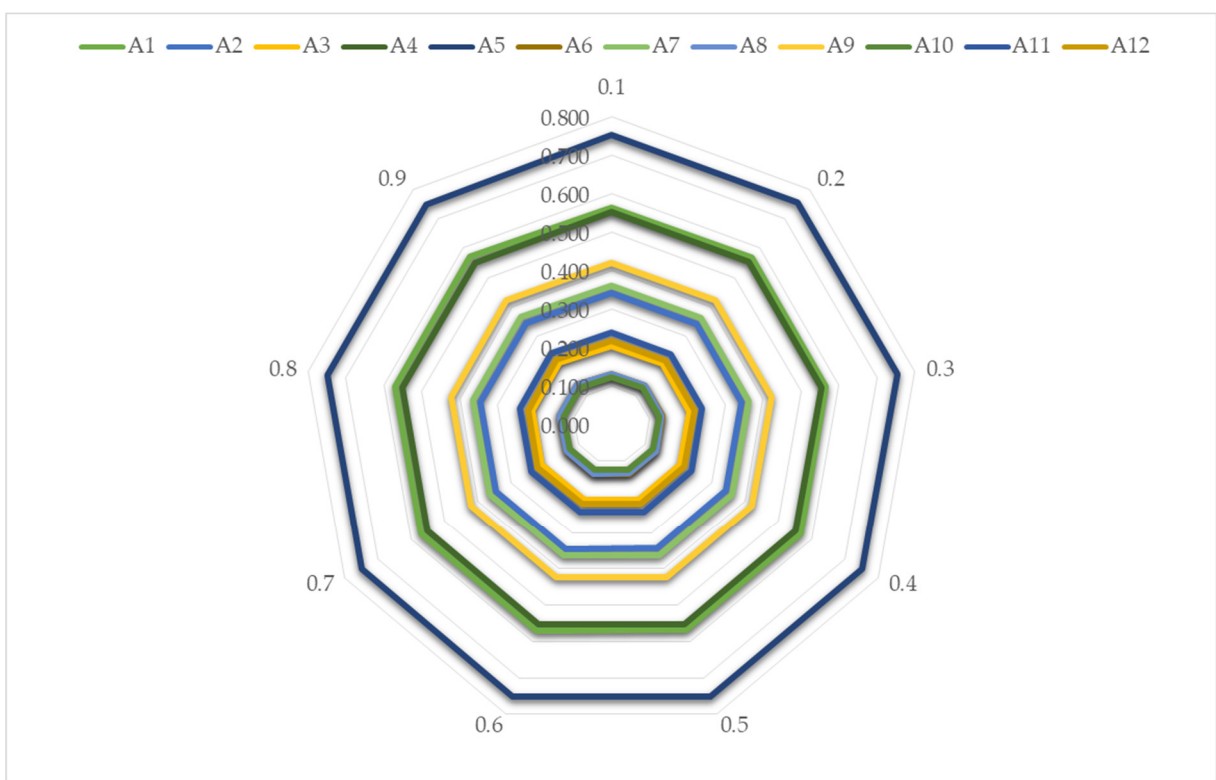

**Figure 5.** Impact of the weights of criteria on ranking priorities.

**Table 13.** Budget allocation according to complete ranking.

| Alternative | Label | TOPSIS Result | Rank | Budget Percentage (%) | Budget Allocation (EUR) |
|---|---|---|---|---|---|
| section 5 | A5 | 0.755 | 1 | 2.67% | 20,413.42 |
| section 1 | A1 | 0.565 | 2 | 3.56% | 27,243.80 |
| section 4 | A4 | 0.554 | 3 | 3.63% | 27,798.92 |
| section 9 | A9 | 0.422 | 4 | 4.77% | 36,514.91 |
| section 7 | A7 | 0.362 | 5 | 5.56% | 42,506.65 |
| section 2 | A2 | 0.342 | 6 | 5.88% | 44,984.55 |
| section 11 | A11 | 0.239 | 7 | 8.41% | 64,331.01 |
| section 12 | A12 | 0.219 | 8 | 9.20% | 70,385.14 |

## 5. Discussion

As construction and maintenance projects of infrastructure elements require the involvement of a large number of stakeholders, it is very important to manage them effectively. The management of stakeholders requires numerous activities to achieve different goals and satisfy the interests of each of them. One way of how this can be achieved is the application of the MCA and MCDM methods in the management process, the purpose of which is multiple. On the one hand, it makes it possible to include the opinions of all stakeholders in the decision-making process, and on the other hand, it makes it possible to break down such a complex problem into its parts, looking at it from different technical, economic, ecological, social, and other aspects.

In this research, the identified stakeholders were divided into three expert groups, which included the users of the analyzed roads, experts on road construction and management, and representatives of the maintenance managers. All of them equally participated in the assessment of the initial relative importance of the criteria that affected the maintenance management of the analyzed roads in the first phase of the AHP method, while in the second phase, the obtained relative weights were adjusted according to the assessment

of the relative weights of the goals provided by the government representatives as the ultimate decision-makers at the highest hierarchical level. However, this was not the case in [27], where a total of 29 experts on road maintenance and renovation participated in determining the relative weights of the defined criteria. There were traffic volume and road condition factors with the greatest relative weights which should have been additionally checked by including other interested participants in the assessment. This is confirmed by a study [13] in which the authors concluded that different types of stakeholders (architects, engineers, managers, and users) determine the weights of criteria and subcriteria differently. Accordingly, the relative weights of the criteria cannot be determined by only one group of stakeholders, but it is necessary to identify and include all the stakeholders who influence the analyzed problem. For example, in this case, the users of the analyzed roads may give priority to the dilapidation and safety of the road, the experts on road construction often see the problem from the aspect of satisfying the technical conditions of the roads, while the representatives of the government and the maintenance managers may see the same problem exclusively from the aspect of budget limitations. By looking at the problem of the maintenance management from only one of the mentioned aspects, it is possible to find a solution for a particular interest group and thereby satisfy the interests of only that group. However, the goal of a sustainable infrastructure maintenance management system is the simultaneous satisfaction of different interests and goals, which implies finding a compromise solution that is acceptable to everyone at the same time. A total of fifteen respondents were included in this research, of which five represented the population of the area in Herzegovina-Neretva County where the analyzed roads were located, five were the experts on road construction and management, and five were the representatives of the Ministry of Transport and Communications of Herzegovina-Neretva County including the Minister of Transport and Communications as the ultimate decision-maker. In this research, an equal number of representatives of each expert group was included to show the differences in viewing the problem from different aspects. However, as Herzegovina-Neretva County is one of the most populated counties in Bosnia and Herzegovina, this problem must surely involve more stakeholders, especially the users of the analyzed roads. Therefore, in future research, it is desirable to include as many interested participants as possible and compare the results.

When it comes to the combination of the two-phase AHP method and the TOPSIS method, on which the proposed multicriteria approach is based, in this paper the classical AHP method included the opinions of a large number of stakeholders, and because of that, it is suitable for solving the problem of infrastructure maintenance management. However, since this research tried to simulate the decision-making process as realistically as possible, the authors note one of the limitations of the classic AHP method. When comparing criteria with relative weights, the opinions of all stakeholders should be taken into account equally. When it comes to the management of stakeholders as one of the areas of knowledge of the management process, it is crucial to include all stakeholders in the decision-making process, but in reality, the final decision, in this case, was made by the representative of the government, i.e., the Ministry of Transport and Communications, who is responsible for the maintenance management of the analyzed roads. To overcome this shortcoming, this research proposed an extension of the AHP method in which the initial relative weights of the criteria determined by all stakeholders were corrected or adapted to the assessment of goals at the highest hierarchical level by the ultimate decision-maker. Thus, an effort was made to create the most effective management system which helped to establish the maintenance management priorities more accurately.

Furthermore, according to [45], another limitation of the AHP method is the high level of subjectivity of the involved stakeholders when comparing the criteria and alternatives. Determining the relative weights of the criteria as well as determining the priority ranking in the classical AHP method comes down mainly to a subjective assessment of the importance of a criterion. To reduce subjectivity in this paper, the AHP method was combined with the TOPSIS method, which was based on the numerical data on the analyzed

roads, which in this case represents an advantage of the TOPSIS method. On the other hand, one of the disadvantages of the TOPSIS method is the impossibility of determining the relative weights of the criteria and the consistency ratio. Furthermore, in [46], the authors determined the best demand management option, comparing the combination of the AHP-PROMETHEE and AHP-TOPSIS methods. The combination of the AHP and TOPSIS methods proved to be a better option for the reason that the relative weights of the criteria can be determined with the AHP method, and by applying the TOPSIS method, it is possible to obtain a complete ranking of the alternatives, while this is not the case with the PROMETHEE method, which requires an additional comparison in pairs. For all of the mentioned reasons, the TOPSIS method was combined with the two-phase AHP method in this paper.

The review of the relevant literature on road infrastructure maintenance indicated a wide application of the MCDM methods, but most of these studies were focused exclusively on determining the weights of the criteria and possibly on ranking alternative solutions. For example, in [27,33], the authors applied the AHP method exclusively to determine the relative weights of the identified criteria. In [29], in addition to determining the relative weights of the criteria, the authors also determined the ranking of maintenance priorities using the AHP method, stating that decisions according to the AHP method are purely subjective. In [31], the authors applied the AHP method to determine the relative weights of the criteria and a combination of the ELECTRE method and the Copeland's technique to determine the priorities of maintenance but without the allocation of financial resources. In [30], the authors also used the AHP method to determine the relative weights of the criteria. Then, using the PROMETHEE II method, the alternatives were ranked according to the maintenance priorities. Subsequently, using the PROMETHEE V method, 11 out of 50 road sections were selected for the first round of investment. However, the results of their research also did not provide the allocation of limited financial resources intended for the maintenance of the analyzed roads. Because of this gap, in this research, the authors tried to obtain the basis for the annual planning of road maintenance management. In this case, the ranking of the management priorities obtained using the TOPSIS method formed a quality basis for the proper allocation of limited financial resources in the function of planning. Based on such allocation, it was possible to design the initial maintenance management plan more effectively. It is well known that in practice, limited financial resources are often distributed without substantiation, which is the most common cause of poor functioning of a management system. However, using the proposed approach, it is possible to ensure a sufficient amount of resources and thereby improve cost management. With all of the above, it is possible to overcome the gaps in the existing literature and thereby improve the function of planning as well as the functions of implementation, monitoring, and control in the maintenance of the analyzed infrastructure elements. In general, in this research, an effort was made to create an effective maintenance management system to preserve the values of the analyzed roads for as long as possible and extend their lifetime.

The TOPSIS method used in this research is based on clear numbers. In future research, it is possible to apply intuitionistic fuzzy sets (IFS) to the maintenance management problem of infrastructural elements according to [25] to increase the accuracy of the decisions made by the involved stakeholders and thus even more realistically simulate the decision-making process with a certain degree of uncertainty. In this case, the TOPSIS method is very suitable because it allows for extensions, and according to [45], it also provides support for interval or fuzzy criteria, interval or fuzzy weights, uncertainty, and lack of information, or vagueness.

## 6. Conclusions

Using the multicriteria analysis (MCA) and selected multicriteria decision-making (MCDM) methods, the process of managing different types of infrastructure systems and their elements can be improved. This paper presents a multicriteria approach to determining the priorities in the maintenance management of roads as the elements of

the selected infrastructure system. With the aim of providing better management of stakeholders, including the identification of the stakeholders and the assessment of their interests, the proposed approach is based on the application of the two-phase AHP method. In contrast to the classic AHP method, which equally takes into account the opinions of all stakeholders, this paper proposes an improved AHP method. In the first phase of the AHP method, the initial relative weights of the criteria were determined with the help of all the stakeholders according to the classic procedure of the AHP method. After that, in the second phase, the initial relative weights of the criteria were adjusted according to the relative weights of the goals determined at the highest level of the hierarchical structure of goals. In this way, the actual decision-making process is simulated in such a way that all involved stakeholders can express their opinions on the analyzed problem, but the final decision is made by the ultimate decision-maker. The final relative weights of the criteria obtained by applying the two-phase AHP method were used as the input data for the TOPSIS method, which was used to determine the priority ranking of the analyzed elements of the selected infrastructure system. The TOPSIS method was chosen due to its perceived advantages over other MCDM methods, which are usually combined with the AHP method but do not provide an option for determining the relative weights of the criteria and the degree of consistency. Based on the distance from the ideal solution, the priority order of the analyzed infrastructure elements was established according to their condition and maintenance needs. Accordingly, the allocation of the limited financial resources intended for the maintenance was made, and the stability of the results obtained using the TOPSIS method was verified by a sensitivity analysis. By applying the proposed multicriteria approach to a concrete case study, it was shown how it is possible to establish a maintenance management concept for the road infrastructure system using a multicriteria analysis, where all stakeholders can be involved in a way that more realistically simulates the decision-making process. Such a concept of maintenance management can be used as the basis for more rational allocation of limited financial resources and help to design an effective maintenance management plan, thus improving the existing management systems. The aforementioned findings provide answers to the research questions posed at the beginning of this paper. According to the literature review, a high level of uncertainty is inherent in the decision-making process concerning the management of infrastructure elements which can be reduced in future research. The proposed multicriteria approach to determining maintenance management priorities can be improved and extended by using intuitionistic fuzzy sets (IFSs). In this way, it is possible to reduce the level of uncertainty and additionally increase the quality of the decision-making process as well as the entire management process of the analyzed infrastructure elements.

**Author Contributions:** Conceptualization, A.B. and N.J.; methodology, A.B. and N.J.; software, A.B. and N.J.; validation, A.B. and N.J.; formal analysis, A.B. and N.J.; investigation, A.B. and N.J.; resources, A.B. and N.J.; data curation, A.B. and N.J.; writing—original draft preparation, A.B. and N.J.; writing—review and editing, A.B. and N.J.; visualization, A.B. and N.J.; supervision, A.B. and N.J.; project administration, A.B. and N.J.; funding acquisition, A.B. and N.J. All authors have read and agreed to the published version of the manuscript.

**Funding:** This research was partially supported through project KK.01.1.1.02.0027, a project co-financed by the Croatian Government and the European Union through the European Regional Development Fund—the Competitiveness and Cohesion Operational Programme.

**Institutional Review Board Statement:** Not applicable.

**Informed Consent Statement:** Informed consent was obtained from all subjects involved in this study.

**Data Availability Statement:** The data presented in this study are available on request from the corresponding author due to restrictions, e.g., privacy or ethical.

**Conflicts of Interest:** The authors declare no conflict of interest.

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
