# Peer review of "Determining Priorities in Infrastructure Management Using Multicriteria Decision Analysis"

_sustainability, doi:10.3390/su152014953_

Round 1
Reviewer 1 Report
The manuscript "Determining priorities in infrastructure management using multi-criteria decision analysis" deals with the problem of decision-making in the management of infrastructure projects with regard to the involved stakeholders. The way to solve this problem using the AHP and TOPSIS methods is presented. The necessary data were collected among three expert groups of prominent stakeholders (users, scientists, and managers). The created priority management system was demonstrated on real road section maintenance projects and demonstrated its applicability. The paper offers an interesting and relevant topic, and the results could be useful for real construction practice. The work is mostly clear, but certain parts should be redefined and described more precisely and in detail.
A list of the main comments and suggestions is provided below.
Abstract
- The Abstract is too general, so it needs to be refined. All important aspects of the work should be mentioned, such as the goal, proposed approach, respondents, novelty, main results, etc.
1. Introduction
- When they appear in the text, define important terms such as infrastructure projects, stakeholders, MCA, and MCDM.
- Full names or abbreviations - This should be written uniformly throughout the work; the first time a term appears, its full name should be written.
- What kind of decisions? Give examples. (line 91)
- Explain why you singled out road maintenance projects as particularly problematic, that is, why they are the area of your interest. (lines 103-104)
- Highlight the research hypothesis/hypotheses.
- The TOPSIS method was not covered in previous research. Expand your references.
- Why did you decide on the AHP and TOPSIS method? Explain.
- The gap in current knowledge needs to be more highlighted. Please point this out.
2. Problem description and methodology
- Explain in more detail the identification and selection of stakeholder groups and their importance and role for road projects. (lines 149-153).
- Explain in more detail the operational, tactical and strategic levels of decision-making, what these levels entail, etc. (lines 154-156).
- It should be described in more detail how the information in Table 1 was created. The content of this table should be commented on extensively.
- Labels EG1, EG2 and EG3 are used inconsistently throughout the work. Please correct.
3. Results
- The procedure of collecting data from respondents needs to be sufficiently covered. How was the data collection carried out, when, from which area were the respondents, what are their characteristics, how many participated in the research, is that number sufficient, etc.?
- Was specific software used during data processing? Name which one.
- Results shown in the tables should be additionally commented on.
- Is there specific general information about the sections of roads that you processed (length, width, year of construction, costs, materials, condition, etc.)? Please show them.
- It is unclear to me how the budget percentages and amounts listed in Table 10 were obtained.
- There is a lack of discussion where the achievement of the goal and hypotheses will be reflected, the advantages and limitations of the research will be highlighted, the results will be compared with other studies, the contribution of the work and novelty will be highlighted, suggestions for future work will be made, etc.
4. Conclusion
- The conclusion acts more like a part of the discussion; it needs to be redefined to write what was actually done in the work and what are the most important findings of this original work.
Literature
The literature is appropriate and relevant, but some more basic and international-level references should be added, according to the comments above.
Author Response
Dear Reviewer,
Please see the attachment.
Sincerely,
Authors

Reviewer 2 Report
The paper presents a multicriteria approach to determining priorities in the infrastructure management using multicriteria analysis.
The topic is object of intensive research in recent years and falls within the scope of the journal.
The abstract is well-written and summarizes the content of the paper.
In the Introduction, the most important notions used in the paper such as infrastructure management, stakeholders, multi-criteria decision making, etc., are briefly mentioned. The Introduction is well-written but some abbreviations are used without explanation. For example, the first time the terms AHP and MCDM appears in the text on line 65 they should not be abbreviated. The reference style used by the authors, for example on line “Eryuruk et al. [7]” is not the one recommended by the journal. I recommend to the authors to stick to the number reference style, i.e. “In [7]”, etc. The TOPSIS method is only mentioned in the last paragraph of the Introduction without explanation. I recommend to the authors to extend the Introduction by briefly describing the TOPSIS and referring to the below source:
Roszkowska, E. Multi-Criteria Decision Making Models by Applying the TOPSIS Method to Crisp and Interval Data. Available online: http://www.mcdm.ue.katowice.pl/files/papers/mcdm11(6)_11.pdf
Furthemore, there are other approaches to the multicriteria decision making which have gained popularity recently and which must be mentioned by the authors. For example, an extension of the fuzzy sets of Lotfi Zadeh, named intuitionistic fuzzy sets, has been applied to multicriteria decision making in the paper:
Atanassov, K. et. al., (2018). Generalized Net Model of Multicriteria Decision Making Procedure Using Intercriteria Analysis. In: Kacprzyk, J., Szmidt, E., Zadrożny, S., Atanassov, K., Krawczak, M. (eds) Advances in Fuzzy Logic and Technology 2017. EUSFLAT IWIFSGN 2017 2017. Advances in Intelligent Systems and Computing, vol 641. Springer, Cham. https://doi.org/10.1007/978-3-319-66830-7_10
In it, an intuitionistic fuzzy approach to mulitcriteria decision making procedure based on intercriteria analysis is proposed. The method proposed in the paper is extremely similar to the one applied by the authors of the present paper and must be cited. Namely, an index matrix the elements of which are intuitionistic fuzzy evaluations of dependencies between the criteria is employed to find out the positive or negative consonance between pairs of criteria.
Another important recent paper, closely related to the topic is
Salimian, S.; Mousavi, S.M.; Tupenaite, L.; Antucheviciene, J. An Integrated Multi-Criteria Decision Model to Select Sustainable Construction Projects under Intuitionistic Fuzzy Conditions. Buildings 2023, 13, 848.
where a new integrated decision analysis model with IFSs. The suggested procedure includes a new decision flow under uncertain situations to define the significance of criteria. In this regard, the weighting of subjective DMs is required for this manner; the only input data needed are an alternative evaluation matrix.
The above sources would provide the required information with regard to the alternative in the modelling of uncertainty in multicriteria decision making.
The paper must be formatted more carefully as page 4 is left half blank.
The quality of Figures 1 and 2 must be improved.
Table 1, 9 etc. have been split on two pages.
The TOPSIS method has been correctly applied.
The conclusions drawn by the authors are supported by the results. I recommend to the authors in their future work to compare the present approach with the one based on intuitionistic fuzzy sets and intercriteria analysis.
Overall, I evaluate highly the paper. I recommend that the paper be published once the authors address adequately my remarks.
Moderate editting of English is required. There are some unclear sentences (see lines 15, 135 and others). Some verbs are used incorrectly.
Author Response

(The authors gave the same response as above.)

Reviewer 3 Report
Thank you for inviting me as a reviewer for the paper. The manuscript is contents fit with the journal’s topics. In this paper, the authors use the AHP - TOPSIS model in the decision-making process related to managing the road infrastructure system.
The authors need to consider the following major points as a limitation or further scope for refining the paper:
. The abstract completely needs to be rewritten. The abstract should include the article's main (1) impact and (2) significance on decision-making systems. Note that a good abstract should contain aims, methods, findings, and recommendations. In addition, it should cover five main elements, introduction, problem statement, methodology, contributions, and results.
. The analysis of the literature must be significantly improved. Authors should present a detailed analysis of previous research related to the research problem and the methods used in the paper. Analyze at least 15-20 recent papers from 2022 and 2023, such as: Badi, I., Abdulshahed, A., & Alghazel, E. (2023). Using Grey-TOPSIS approach for solar farm location selection in Libya. Reports in Mechanical Engineering, 4(1), 80–89; Hussain, O.A.I.; Moehler, R.C.; Walsh, S.D.C.; Ahiaga-Dagbui, D.D. Minimizing Cost Overrun in Rail Projects through 5D-BIM: A Systematic Literature Review. Infrastructures 2023, 8, 93.; Jagtap, M., & Karande, P. (2023). The m-polar fuzzy set ELECTRE-I with revised Simos’ and AHP weight calculation methods for selection of non-traditional machining processes. Decision Making: Applications in Management and Engineering, 6(1), 240–281. Based on literature analysis the authors need to discuss their contributions compared to those in related papers. The research gap and motivation should be clarified in the introduction section. Authors should begin with the problem, the gap, then propose the research question, and just after that say what they want to do to address that. Where is the gap? And you should clearly why it is a gap? Once again, if you say that it is a gap, then try to build a case for the gap.
. Reduce the number of self-citations. Almost a third of the papers are self-citations.
. At the end of the introduction, announce the rest sections of the paper in one paragraph.
. From the author's explanation, it is not clear what the second phase of the AHP method is. I believe that this is completely clear to the authors, but it should be clear to others as well.
. You are reorganizing the paper: Introduction and literature analysis; Description of methods; Case study; Sensitivity analysis and comparison with other methods; Discussion and conclusion.
. The criteria should be described in detail.
. A sensitivity analysis is missing. A sensitivity analysis should be done using the change in weighting coefficients.
. What about comparing the results with other methods?
. There is a lack of serious discussion of the results and definition of managerial implications.
. What are the limitations of the model?
General observation and recommendations:
- Show the methods step by step in the description and later in the case study.
- The case study must be well described because this is where the contributions of the paper lie since the applied methods are quite well known.
Author Response

(The authors gave the same response as above.)

Round 2
Reviewer 1 Report
Dear authors,
Thank you for your effort and adequate response to my remarks.
I have no comments on the content. I think the MS has improved significantly compared to the first version. I want to congratulate you on your work and wish you luck in the future.
I'm just asking you to review MS again and correct any editing errors. For example, correct the errors in the lines:
290 - instead of Figure 2, it should be Figure 1
434 - 1 is the best condition, and 10 is also the best condition?
439 - appeared unnecessary s
519 - instead of Table 3, Table 2 should be written
831- is not Tijanic-Štrok but Tijanić Štrok.
Also, the reference to pictures and tables is inconsistent, sometimes with a dot and sometimes without a dot.
Author Response
Dear Reviewer,
all your suggestions from the 2nd round of the review have been corrected by the authors in the new version of this manuscript. Accordingly, detailed explanations can be found in the attachment.
We hope that the new version of this manuscript will now be at a satisfactory level for acceptance.
Sincerely,
Authors

Reviewer 2 Report
Thank you for revising the paper. The paper quality has been improved. I recommend that the paper be published in the present form.
Minor editing is required. There are some style and grammar mistakes.
Author Response
Dear Reviewer,
thank you for accepting our paper as well as for the time and effort you invested to raise the quality of this paper to a satisfactory level for publication in the Sustainability journal.
The authors also made the requested style and grammar corrections to the text of this manuscript, and if you see any other mistakes, please point them out to us.
Sincerely,
Authors
Reviewer 3 Report
All the reviewers' comments have been addressed carefully and sufficiently. The revisions are rational from my point of view. I think the current version of the paper can be accepted
Author Response
Dear Reviewer,
thank you for accepting our paper as well as for the time and effort you invested to raise the quality of this paper at a satisfactory level for publication in Sustainability journal.
Sincerely,
Authors